# Nonlinear expression patterns and multiple shifts in gene network interactions underlie robust phenotypic change in *Drosophila melanogaster* selected for night sleep duration

**Caetano Souto-Maior**[¤]*, **Yazmin L. Serrano Negron**, **Susan T. Harbison***

Laboratory of Systems Genetics, Systems Biology Center, National Heart Lung and Blood Institute, Bethesda, Maryland, United States of America

¤Current address: Basque Center for Applied Mathematics (BCAM), Bilbao, Spain
* caetanosoutomaior@proton.me (CS-M); susan.harbison@nih.gov (STH)

**Data Availability Statement:** Night sleep phenotypes per selection scheme/sex/generation/

## Abstract

All but the simplest phenotypes are believed to result from interactions between two or more genes forming complex networks of gene regulation. Sleep is a complex trait known to depend on the system of feedback loops of the circadian clock, and on many other genes; however, the main components regulating the phenotype and how they interact remain an unsolved puzzle. Genomic and transcriptomic data may well provide part of the answer, but a full account requires a suitable quantitative framework. Here we conducted an artificial selection experiment for sleep duration with RNA-seq data acquired each generation. The phenotypic results are robust across replicates and previous experiments, and the transcription data provides a high-resolution, time-course data set for the evolution of sleep-related gene expression. In addition to a Hierarchical Generalized Linear Model analysis of differential expression that accounts for experimental replicates we develop a flexible Gaussian Process model that estimates interactions between genes. 145 gene pairs are found to have interactions that are different from controls. Our method appears to be not only more specific than standard correlation metrics but also more sensitive, finding correlations not significant by other methods. Statistical predictions were compared to experimental data from public databases on gene interactions. Mutations of candidate genes implicated by our results affected night sleep, and gene expression profiles largely met predicted gene-gene interactions.

## Author summary

Understanding the molecular bases of phenotypes remains a challenge of complex trait biology. We used a combination of selective breeding, RNA-Seq, and Gaussian Process modeling to determine whether *de novo* gene expression networks could be derived for sleep duration in *Drosophila*. We bred flies with long and short sleep times, and sequenced RNA from the flies at each generation of selection. Using a hierarchical Bayesian

population replicate are listed in S15 Table. All scripts used for the model have been deposited in Git Hub https://github.com/caesoma/Multiple-shifts-in-gene-network-interactions-shape-phenotypes-of-Drosophila-melanogaster. All RNA-Seq data from this study are available from the National Center for Biotechnology Information (NCBI) Gene Expression Omnibus (GEO) under the accession number GSE202600.

**Funding:** This research was supported by the Intramural Research Program of the National Institutes of Health, the National Heart Lung and Blood Institute, ZIA HL006146 to STH. CSM, YLSN, and STH received salary support from the NIH funding listed above. The funders had no role in study design, data collection and analysis, decision to publish, or preparation of the manuscript.

**Competing interests:** The authors declare that they have no conflict of interest.

Generalized Linear Model, we identified genes with altered expression across generation in the selected populations. Gene expression trajectories were largely non-linear across time, however, so we developed a Gaussian Process method to more accurately model the data. The Gaussian Process provides an adaptable framework that adjusts to the complexity of gene expression patterns we observed, eliminating the need to specify or assume a specific polynomial model. The Gaussian Process also enabled us to compute covariances among pairs of genes, elucidating gene expression networks for sleep duration. Follow-up mutational analyses validated the candidate genes' effects on sleep duration and transcriptional analyses of the mutations largely confirmed gene expression network predictions. The Gaussian Process framework is broadly applicable to gene expression data collected across time.

## Introduction

Despite the plethora of modern and increasingly refined molecular biology assays—from DNA to metabolites and beyond—systematically uncovering the molecular bases of phenotypes remains one of the thorniest challenges in biology. "Omics" approaches allow whole genome, transcriptome, proteome, and other "omes" to be generated and candidate genes to be fished out of these high dimensional data, but understanding how these biomolecules interact even in the simplest pathways requires painstaking follow-on experimentation, construction of databases, and an immense collective effort to make connections from disjointed assays into a coherent model. Despite the large amount of studies and data generated for many systems, a full understanding of underlying processes has not yet been achieved; this is clear indication that better methods are needed to obtain the understanding of biological processes from data. For complex traits the task is even more difficult. Sleep is a complex phenotype the evolution of which remains a classic mystery in biology. Although sleep and sleep-like behavior is conserved among species, its main purpose is not completely understood, and hypotheses for its purpose span functions like conservation of resources [1–3], pruning of synapses and memory formation [4–7], and management of metabolite and waste products [8, 9]. It is plausible that sleep is a manifestation of multiple functions, and that it involves the activity of many genes to regulate a complex higher-level function; indeed many genes have been implicated in sleep [10–20]. Assuming anything but the simplest possible model would therefore require a description that accounts for this complexity in the interactions of genes and gene products.

Artificial selection plus sequencing/resequencing is a powerful approach for identifying heritable variation in phenotypes and their underlying molecular bases [21], typically assaying DNA or RNA expression in the initial and evolved populations and comparing them to controls [22, 23]. Coupling selection with gene expression identified candidate genes for diurnal preference [24], olfactory behavior [25, 26], food consumption [27], mating behavior [28], resistance to parasitism [29], environmental stressors [30, 31], ethanol tolerance [32], and aggressive behavior [33]. Caveats of that method include often not having molecular data on the intermediate generations, and relying on traditional statistical methods to assess the significance of polymorphic variants. In the case of gene expression, RNA levels are often modeled for each gene individually using linear models, without further consideration of the processes involved or interactions between genes. Inferring interaction between genes (as opposed to individual changes) requires observations of how the genes covary in time. Correlation [34, 35] or information theory-based methods (and others, reviewed in [36–38]) could be applied

to estimate the relationship between the genes when that information is present, but neither is time course data usually available, nor are these methods standard in artificial selection experiments.

Recent work applies Gaussian Process models [39, 40] to data sampled over time in order to evaluate dynamic parameters. Where gene and protein expression dynamics can be modeled with differential equations, Gaussian Process methods estimate parameters for non-linear systems [41], elucidate the spatial-temporal dynamics of developmental morphogen gradients [42], model signaling and gene regulatory networks [43, 44] infer latent transcription factor activity [45], and find transcription factor targets [46]. Other applications of Gaussian Processes account for missing or irregularly sampled gene expression data [47], model spatial interactions between cells [48] and generate clusters [47, 49]. Gaussian Processes can also infer relationships among multiple disparate data types [50, 51] to explore, for example latent processes underlying spatial-temporal relationships among brain structure, brain activity metrics, and behavior [52], or relationships among multi-modal spatially- and temporally-varying data [53]. Gaussian Processes thus provide a flexible framework for the estimation of latent relationships.

In this work we have artificially selected *Drosophila melanogaster* for increased or decreased night sleep duration and sequenced the mRNA of the flies from each generation of selection. The selection procedure produced both long- and short-sleeping fly populations significantly deviant from unselected controls. The RNA sequence data, which consisted of expression levels as a function of time (measured in generations), was analyzed using a Multi-Channel Gaussian Process [50, 51] where each gene is described by one of these "channels", and their relationships are estimated by an underlying covariance structure in the model. We describe the expression of 85 genes that had significant changes in the artificial selection long or short schemes along generation common to both males and females. We used this model to infer the magnitude of all 3,570 possible pairwise interactions between all possible pairs of genes. Results from this analysis and comparison to unselected controls suggest that multiple shifts in interactions underlie the increase and decrease of night sleep duration, with 145 interactions not being observed in the controls. Further experiments revealed candidate genes that impact night sleep and confirm these interactions.

## Materials and methods

### Construction of outbred population

We constructed an outbred population of flies using ten lines from the *Drosophila* Genetic Reference Panel (DGRP) [54, 55] with extreme night sleep phenotypes [11]. Five lines had the shortest average night sleep for both males and females combined in the population: DGRP_38, DGRP_310, DGRP_365, DGRP_808, and DGRP_832. The other five lines had the longest average night sleep in the population: DGRP_235, DGRP_313, DGRP_335, DGRP_338, and DGRP_379. The ten lines were crossed in a full diallel design, resulting in 100 crosses. Two virgin females and two males from the F1 of each cross were randomly assigned into 20 bottles, with 10 males and 10 females placed in each bottle. At each subsequent generation, 20 virgin females and 20 males from each bottle were randomly mixed across bottles to propagate the next generation. The census population size was 800 for each generation of random mating. This mating scheme was continued for 21 generations, resulting in the Sleep Advanced Intercross Population, or SAIP [10, 56]. The SAIP was maintained by pooling the flies from each bottle together, then randomly assigning 20 males and 20 females to each bottle each generation.

## Artificial selection procedure for night sleep

At generation 47 of the SAIP, we began the artificial selection procedure, which we defined as generation 0. We seeded six bottles with 25 males and 25 females mixed from all bottles of the outbred population. Two replicate bottles were designated for the short-sleeping protocol (S1 and S2), two for the long-sleeping protocol (L1 and L2), and two for a control (unselected) protocol (C1 and C2). Each generation, 100 virgin males and 100 virgin females were collected from each of the six population bottles. Virgins were maintained at 20 individuals to a same-sex vial for four days to control for the potential effects of social exposure on sleep [57]. Flies were placed into Trikinetics (Waltham, MA) sleep monitors, and sleep and activity were recorded continuously for four days. We used an in-house C# program (R. Sean Barnes, personal communication) to calculate sleep duration, bout number, and average bout length during the night and day, as well as waking activity. We also calculated sleep latency, defined as the number of minutes prior to the first sleep bout after the incubator lights turn off. In addition, we computed the coefficient of environmental variation ($CV_E$) for each sleep trait as the product of the standard deviation in each replicate population ($\sigma$) divided by the mean ($\mu$) $\times$ 100 [58].

All sleep traits including night sleep duration were averaged over the four-day period. For the short (long)-sleeping populations, we chose the 25 males and 25 females in each replicate population having the lowest (highest) average night sleep as parents for the next generation. Any flies found dead were discarded, and the next shortest (longest)-sleeping fly was used in order to ensure that 25 females and 25 males were used as parents. For the control populations, we chose 25 males and 25 females at random to start the next generation. Flies were not mixed across replicate populations. We repeated this procedure for 13 generations.

## Quantitative genetic analyses of selected and correlated phenotypic responses

We analyzed the differences in night sleep among selection populations as well as other potentially correlated sleep traits using a mixed analysis of variance (ANOVA) model:

$$Y = \mu + Sel + Rep(Sel) + Sex + Gen + Sel \times Sex + Sel \times Gen + Rep(Sel) \times Sex$$
$$+ Rep(Sel) \times Gen + Sex \times Gen + Sel \times Sex \times Gen + Rep(Sel) \times Sex \times Gen + \varepsilon$$

where $Y$ is the phenotype; $\mu$ is the overall phenotypic mean; *Sel*, *Sex*, and *Gen* are the fixed effects of selection scheme (short- or long-sleeper), sex, and generation, respectively; *Rep* is the random effect of replicate population; and $\varepsilon$ is the error term. The $CV_E$ traits were assessed using the same model with the replicate terms removed. A statistically significant *Sel* term indicates a response of the trait to selection for night sleep; a significant *Sel* $\times$ *Sex* term indicates a sex-specific response to selection. We repeated the analysis for sexes separately using the reduced model

$$Y = \mu + Sel + Rep(Sel) + Gen + Sel \times Gen + Rep(Sel) \times Gen + \varepsilon$$

where the terms are as defined above. We also analyzed the response to selection in each generation separately using the reduced model

$$Y = \mu + Sel + Rep(Sel) + Sex + Sel \times Sex + Rep(Sel) \times Sex + \varepsilon$$

and the reduced model

$$Y = \mu + Sel + Rep(Sel) + \varepsilon$$

for each sex separately per generation. Finally, we analyzed the change in sleep parameters over generations in the control populations using the model

$$Y = \mu + Rep + Sex + Gen + Rep \times Sex + Rep \times Gen + Sex \times Gen + Rep \times Sex \times Gen + \varepsilon$$

where each factor is as defined above. We estimated realized heritability ($h^2$) using the breeder's equation:

$$h^2 = \frac{\Sigma R}{\Sigma S}$$

where $\Sigma R$ and $\Sigma S$ are the cumulative selection response and differential, respectively [59]. The selection response is computed as the difference between the offspring mean night sleep and the mean night sleep of the parental generation. The selection differential is the difference between the mean night sleep of the selected parents and the mean night sleep of the parental generation.

## RNA extraction and sequencing

As described above, sleep was monitored in 100 virgin males and 100 virgin females each generation. Twenty-five flies of either sex were used as parents for the next generation, leaving 75 flies of each sex in each selection and control population. Four pools of 10 flies of each sex were chosen at random from these 75 flies and frozen for RNA extraction at 12:00 pm (i.e., ZT6). This timepoint was arbitrarily chosen and is during the fly's active period. RNA was extracted from two of these pools; the remaining two pools were kept as back-up samples and used if needed. Samples were collected for the initial generation (0), and all subsequent generations. RNA was extracted using Qiazol (Qiagen, Hilden, Germany), followed by phenol-chloroform extraction, isopropanol precipitation, and DNase digestion (Qiagen, Hilden, Germany). Qiagen RNeasy MinElute Cleanup kits (Qiagen, Hilden, Germany) were used to purify RNA according to the manufacturer's instructions. With the exception of generation 1, which had RNA that was degraded, RNA from all other generations was sequenced. This produced 312 RNA samples (6 populations × 13 generations × 2 sexes × 2 replicate RNA samples).

Poly-A selected stranded mRNA libraries were constructed from 1 $\mu g$ total RNA using the Illumina TruSeq Stranded mRNA Sample Prep Kits (Illumina, San Diego, CA) according to manufacturer's instructions with the following exception: PCR amplification was performed for 10 cycles rather than 15 in order to minimize the risk of over-amplification. Unique barcode adapters were applied to each library. Libraries were pooled for sequencing. The pooled libraries were sequenced on multiple lanes of an Illumina HiSeq2500 using version 4 chemistry to achieve a minimum of 38 million 126 base read pairs. The sequences were processed using RTA version 1.18.64 and CASAVA 1.8.2.

## RNA alignment of reads

Sequences were assessed for standard quality parameters using fastqc (0.11.4) (Babraham Institute, Cambridge, UK). Reads were aligned to the FB2015_04 Release 6.07 reference annotation of the *Drosophila melanogaster* genome using STAR [60]. Default parameters were used except that the minimum intron size was specified as 2, and the maximum intron size was specified as 268,107, consistent with the largest intron size in the *D. melanogaster* genome. STAR outputs aligned sequence to a SAM file format, which contains the code '*NH*' [60]. An *NH* of 1 indicates a uniquely mapped read, while $NH > 1$ indicates that the read did not map uniquely. HTSeq was used to count only the uniquely mapped reads ($NH = 1$) [61].

## Principal Component Analysis (PCA)

It was expected from previous studies of gene expression that there would be large differences in gene expression due to sex [62–72]. We performed Principal Component Analysis to assess those differences (S1 Fig). The principal components of the normalized RNA-seq count normalized matrix were computed, with each gene being treated as a different variable, and each sample a different observation. Samples were projected in the planes of the ten first components, and clustering according to the experimental labels was inspected visually.

## Gene normalization and filtering

The combined genic and intergenic counts were normalized by the expression of a pseudo-reference sample computed from the geometric mean of all samples, using the method described by Love *et al.* [73]. Filtering was performed by computing the 95th percentile of the distribution of normalized, base 2 logarithm, levels in the intergenic regions for males and females and using those values as cut-off level for the genic regions—i.e. any genes that did not have expression above this level for at least one sample were removed from further analyses [74]. The (linear scale) cutoff expression value for males was 48.6, and for females 102.

## Generalized Linear Model analysis of expression data

Analysis of differential expression between selection schemes was initially performed for each gene independently. Given the separation of the expression levels by sex seen in the PCA analysis, analyses were conducted separately for the subsets of male or female flies. We implemented a generalized linear model (GLM) with a hierarchical structure to account for non-independent, replicate-specific parameters. The description is similar to a generalized linear mixed model (GLMM), but uses a Bayesian formulation to specify the hyper-priors and is fully described below. Normalization factors for the RNA levels was performed using the scheme described by Love *et al.* [73]. A negative binomial likelihood was used and parameterized with the mean (given by the prediction of the linear model) and dispersion parameters; the number of samples (156 for each sex) allowed estimation of the latter together with model coefficients, dispensing with the need of other schemes applied when the number of samples is small, commonly implemented in some packages.

Bayesian inference was used and parameter priors were exploited to treat replicate effects in a hierarchical formulation [75]. Specifically, for each replicate-dependent parameter (say $\beta_{short,rep}$), two parameters were specified at the top-level ($\mu_{short}$ and $\sigma_{short}$), given (hyper-)priors, and estimated from the data together with all other parameters. Below that, both replicate-specific model parameters ($\beta_{short,1}$ and $\beta_{short,2}$) are given the same gaussian prior using top-level parameters (e.g. $\beta_{short,1} \sim \mathcal{N}(\mu_{short}, \sigma_{short})$ for that coefficient in replicate 1 as well as replicate 2). Under this formulation the full model for the expression of a gene *j* is given by $log\mu_j \propto sel_{rep} + gen + sel \times gen_{rep}$, where a relationship between each set of replicate-dependent parameters is enforced hierarchically through their higher level common parameters and hyperpriors. Explicitly, we have:

$$
\begin{aligned}
\eta_j \quad &= log\mu_j \\
&= [\beta_1, \beta_2, \beta_{short,1}, \beta_{short,2}, \beta_{long,1}, \beta_{long,2}, \beta_{gen}, \beta_{short \times gen,1}, \beta_{short \times gen,2}, \\
&\quad \beta_{long \times gen,1}, \beta_{long \times gen,2}]X
\end{aligned}
$$

where *X* is the design matrix, with binary 0/1 variables indicating parameters that apply to specific treatments (e.g. the entries multiplying $\beta_1, \beta_2$, are present for all, that $\beta_{short,1}$, is present for short sleepers from replicate 1, etc.) except for parameters dependent on the *gen* variable

**Table 1. Parameter names, description, design values, and priors for Bayesian inference.**

| Parameter | Description | Design values | Prior |
|---|---|---|---|
| $\mu_{control}$ | Hyperprior on mean of $\beta_{rep}$ | n/a | $\mathcal{N}(\bar{y}_0, 1)$ |
| $\sigma_{control}$ | Hyperprior on (square root of) variance of $\beta_{rep}$ | n/a | $Cauchy(0, 1)$ |
| $\mu_{short}$ | Hyperprior on mean of $\beta_{short,rep}$ | n/a | $\mathcal{N}(0, 1)$ |
| $\sigma_{short}$ | Hyperprior on variance of $\beta_{short,rep}$ | n/a | $Cauchy(0, 1)$ |
| $\mu_{long}$ | Hyperprior on mean of $\beta_{long,rep}$ | n/a | $\mathcal{N}(0, 1)$ |
| $\sigma_{long}$ | Hyperprior on variance of $\beta_{long,rep}$ | n/a | $Cauchy(0, 1)$ |
| $\mu_{short\times gen}$ | Hyperprior on mean of $\beta_{short\times gen,rep}$ | n/a | $\mathcal{N}(0, 1)$ |
| $\sigma_{short\times gen}$ | Hyperprior on variance of $\beta_{short\times gen,rep}$ | n/a | $Cauchy(0, 1)$ |
| $\mu_{long\times gen}$ | Hyperprior on mean of $\beta_{long\times gen,rep}$ | n/a | $\mathcal{N}(0, 1)$ |
| $\sigma_{long\times gen}$ | Hyperprior on variance of $\beta_{long\times gen,rep}$ | n/a | $Cauchy(0, 1)$ |
| $\beta_1$ | Intercept for replicate 1 | 0, 1 | $\mathcal{N}(\mu_{control}, \sigma_{control})$ |
| $\beta_2$ | Intercept for replicate 2 | 0, 1 | $\mathcal{N}(\mu_{control}, \sigma_{control})$ |
| $\beta_{short,1}$ | Effect from short sleeper, replicate 1 treatment | 0, 1 | $\mathcal{N}(\mu_{short}, \sigma_{short})$ |
| $\beta_{short,2}$ | Effect from short sleeper, replicate 2 treatment | 0, 1 | $\mathcal{N}(\mu_{short}, \sigma_{short})$ |
| $\beta_{long,1}$ | Effect from long sleeper, replicate 1 treatment | 0, 1 | $\mathcal{N}(\mu_{long}, \sigma_{long})$ |
| $\beta_{long,2}$ | Effect from long sleeper, replicate 2 treatment | 0, 1 | $\mathcal{N}(\mu_{long}, \sigma_{long})$ |
| $\beta_{gen}$ | Treatment-independent generation effect | 0–13 | $\mathcal{N}(0, 2)$ |
| $\beta_{short\times gen,1}$ | Interaction short by generation, rep 1 effect | 0–13 | $\mathcal{N}(\mu_{short\times gen}, \sigma_{short\times gen})$ |
| $\beta_{short\times gen,2}$ | Interaction short by generation, rep 2 effect | 0–13 | $\mathcal{N}(\mu_{short\times gen}, \sigma_{short\times gen})$ |
| $\beta_{long\times gen,1}$ | Interaction long by generation, rep 1 effect | 0–13 | $\mathcal{N}(\mu_{long\times gen}, \sigma_{long\times gen})$ |
| $\beta_{long\times gen,2}$ | Interaction long by generation, rep 2 effect | 0–13 | $\mathcal{N}(\mu_{long\times gen}, \sigma_{long\times gen})$ |
| $\alpha$ | Negative binomial dispersion | n/a | $Uniform(0, 10^9)$ |

$\bar{y}_0$ denotes the mean expression of all samples at generation zero.

which takes the value of the generation (e.g. 0 through 13 for the entries multiplying the $\beta_{gen}$ parameter in all treatments, and for those multiplying $\beta_{short\times gen,1}$ for short sleepers from replicate 1, etc.). Table 1 lists all parameters, their descriptions, design matrix values associated to them, and priors.

Maximum a posteriori probability (MAP) estimates and confidence intervals were obtained using the Stan package [76]. Significance was calculated using a likelihood ratio test comparing the point estimates from the full model to a reduced model not including the interaction terms (i.e. $log\mu_{j,rep} = sel_{rep} + gen$). Model *p*-values were corrected for multiple testing using the Benjamini-Hochberg method [77], with significance defined at the 0.001 level, consistent with the lower threshold applied in other artificial selection studies [28, 32, 33].

## Calculation of non-parametric correlations between genes

The correlation coefficients ($\rho$) between any two pairs of genes can be computed directly from the data. Pearson correlation assumes the relationship between the two variables is linear, while Spearman correlation is rank-based and therefore accommodates non-linear relationships, although it still assumes the relationship is monotonically increasing or decreasing. We therefore computed Spearman correlations between genes that were found to be significant for both males and females in the GLM analysis—one correlation coefficient was obtained for the data subset from each sex-selection combination. The significance of each correlation coefficient is tested using the null hypothesis that $\rho = 0$. Because the main interest is the interaction

between genes in the selected populations that are different from controls we compare the coefficients by computing and comparing the confidence intervals for $\rho_{sel}$ (where *sel* can be "short" or "long") and $\rho_{control}$ using the normal approximation to $arctanh(\rho)$ [78]. We note that this is not exactly equivalent to the significance testing of the null hypothesis that $\rho_{sel} = \rho_{control}$ [79] (which relies on computing the confidence interval for $\rho_{sel} - \rho_{control}$ using the same method), since it overestimates the total variance (i.e., one would find fewer significant instances). Nevertheless, the approach is valid and is more broadly applicable, in that it can be computed when a joint distribution with the two variables cannot be obtained—we use the term "significant" for either kind of difference, but explicitly state which one is used.

## Gaussian Process regression

Gaussian Processes (GP) are an alternative function-space formulation to the well-known weight-space linear models of the form $y = f(x) + \varepsilon$; their use dates back to the 19th century and they have been covered extensively in the statistical and information theory literature [80], becoming popular in machine learning applications [39, 81], and more recently implemented in less technical contexts like the life sciences [40]. We give a brief overview of their usefulness, motivate their use in this work, and point to the references above for formal description of the method.

The weight-space linear model expresses the observations in terms of explicit linear coefficients (or weights) of the independent variable, $x$, possibly with further basis function expansions (e.g. square, $x^2$, or higher order polynomials, $x^n$), for instance $y = \beta_0 + \beta_1 x + \beta_2 x^2 + \varepsilon$, (where $\varepsilon$ is normally distributed noise). Gaussian Processes describe the basis functions implicitly instead, with $y \sim \mathcal{N}(\mu, K)$; that is, a set $y$ of $N$ observations is distributed according to a multivariate normal distribution with mean given by the vector $\mu$ (of size $N$) and covariance between the values of $x$ given by the matrix $K$ (with dimension $N \times N$). The entries of this matrix in row $i$, column $j$ are defined by some covariance function such that $k_{ij} = cov(x_i, x_j)$—if the covariance function is linear in the values of $x$, for instance, the prediction for $y$ is a straight line similar to $y = \beta_0 + \beta_1 x$. Formulating the model in terms of function-space enables the use of flexible sets of basis functions; this approach of only implicitly describing a basis function, thus avoiding specification of a potentially large basis is called the "kernel trick". Functions like the commonly used squared exponential kernel can be shown to be equivalent to an infinite number of basis functions [39], and therefore cannot be incorporated in the explicit terms of the weight-space formulation.

While Gaussian Processes are a classic formulation in statistics, the recent surge in machine learning applications has popularized its use in the natural sciences. They have been used to analyze gene expression by using their flexible output in combination with ordinary differential equations [41, 43, 44, 46], with clustering approaches [49], within other regression models [82], or modeling spatial covariance [48]. In the context of our experimental design Gaussian Process Regression could be used as a flexible alternative to GLMs, with each selection scheme having a different mean function $\mu_{sel}$ and a squared exponential covariance function $k(x, x') = \sigma_f^2 c(x, x') = \sigma_f^2 \exp\left(\frac{|x-x'|^2}{2l^2}\right)$ where $x$ takes the values of the generations in our experiment. The exponentiated term gives the correlation $c(x, x')$ between a pair of time points, with parameter $\ell$ modulating the correlation level given a distance $r = x - x'$, and $\sigma_f^2$ being the signal variance of the data. Under this model, unlike with the GLM analysis, the change in RNA-seq counts is a function not of slope coefficients but of the signal variance $\sigma_f^2$. It is worth noting that the signal variance is a scalar constant for all terms in the covariance matrix, so it can also be written

as $K = \sigma_f^2 C$, where $C$ is analogous to $K$ but with correlations instead of covariances, a notation that will be useful shortly.

## Multi-channel Gaussian Processes

Despite the extensive use of Gaussian Processes, most applications in the life sciences have been restricted to single-channel GPs; that is, models that only describe one set of observations at a time (here the expression time series for a single gene). These models—in this aspect not unlike GLMs—describe expression of genes independently, i.e. they implicitly assume genes do not interact in any way. Gaussian Processes can however be extended to include covariance between two or more sets of observations, a formulation that seems to be underexploited in the biological literature (but see [53] and [52]). The different dependent variables $y_i$ are sometimes called channels or tasks, and the resulting model is called a multi-task or multi-channel Gaussian Process. The details of the specification of this model can be found in [51] and [50], which we summarize below. For an array of two genes only, for instance, instead of describing each vector $y_1$ and $y_2$ separately as multivariate gaussians of dimension $N_1$ and $N_2$, respectively, the concatenated vector $[y_1 \, y_2]^T$ with $N_1 + N_2$ observations can be modeled as a single multivariate gaussian with a covariance matrix of $K$ dimensions $(N_1 + N_2) \times (N_1 + N_2)$, or $[y_1 \, y_2]^T \sim \mathcal{N}(\mu, K)$. The diagonal blocks of the covariance matrix with dimensions $N_1 \times N_1$ and $N_2 \times N_2$ are the same as above, and the off-diagonal blocks of dimensions $N_2 \times N_1$ and $N_1 \times N_2$ specify the correlations $c_{12ij}(x_{1i}, x_{2j}) = \exp\left(\frac{|x_{1i} - x_{2j}|^2}{\ell_1^2 + \ell_2^2}\right)$ between the two points $ij$ from channels 1 and 2 [50].

Finally, the signal variance for each of those blocks need to be specified, and the final matrix is given by $K = \begin{bmatrix} K_{11} & K_{12} \\ K_{21} & K_{22} \end{bmatrix} = \begin{bmatrix} \sigma_1^2 C_{11} & \sigma_{12}^2 C_{12} \\ \sigma_{12}^2 C_{12} & \sigma_2^2 C_{22} \end{bmatrix}$ [51], and the mean of the multivariate gaussian is specified by a concatenated vector $\mu = [\mu_1 \, \mu_2]^T$. The number of parameters is reduced by recognizing that the covariance matrix is symmetric so in this example $\sigma_{21}^2 = \sigma_{12}^2$, where we also dropped the subscript *f*. For this model, the variation in the RNA levels of say gene 1 is a function not only of $\sigma_1^2$, but also of $\sigma_{21}^2 = \sigma_{12}^2$. Therefore, fitting the data with this model infers interaction between genes from scratch without any external information not contained in the array of RNA-seq counts.

The model can be extended to any number of genes, although computational requirements for performing the necessary matrix operations on $K$ also grow with its size and may be limiting—the computational and mathematical limitations of this approach are discussed in S1 Appendix.

## Bayesian MCMC inference of Gaussian Processes

Analogously to GLM models, we maintain the negative binomial likelihood for the Gaussian Process inference, but unlike the transition between linear models and their generalized versions, the incorporation of non-gaussian likelihoods is not as straightforward, and requires methods to approximate the underlying latent Gaussian Process model, leading to what is sometimes referred to as Gaussian Process Classification [39]. Because of the Bayesian inference implemented for this model we chose to infer the latent function via Markov Chain Monte Carlo sampling as these variables can be estimated jointly with the other parameters and have priors that by design are standard gaussian, and therefore are straightforward to specify. Table 2 gives the description of all parameters in the Multi-Channel Gaussian Process model and their priors.

**Table 2. Parameter names, description, and priors for Gaussian Process Bayesian inference.**

| Parameter | Description | Prior |
|---|---|---|
| $s$ | Standard deviations of data (one for each channel) | n/a |
| $\hat{\sigma}_i^2, (V_{\sigma,i})$ | Signal variance expectation (variance) from single-channel $i$ model | n/a |
| $\hat{\ell}_i, (V_{\ell,i})$ | Bandwidth expectation (variance) from single-channel $i$ model | n/a |
| $\sigma_{ii}^2$ | Signal variance for channels $i$ | $\mathcal{N}(\hat{\sigma}_i, \sqrt{V_{\sigma,i}})$ |
| $\sigma_{ij}^2$ | Signal covariance between channels $i$ and $j$ | $\mathcal{N}(0, max(s))$ |
| $\ell$ | Bandwidth parameters | $\mathcal{N}(\hat{\ell}_i, \sqrt{V_{\ell,i}})$ |
| $\tilde{f}$ | Gaussian Process latent normal variates | $\mathcal{N}(0,1)$ |
| $\phi$ | inverse of square of dispersion parameter ($\phi = 1/\alpha^2$) | $\mathcal{N}(0,1)$ |

The number of covariance parameters in a multi-channel Gaussian Process model with *M* channels is $(M^2 - M)/2$, and the total number of parameters scales roughly as $\mathcal{O}(M^2)$ as the number of channels becomes large. For 100 genes, for instance, that would result in about 5,000 covariances. Due to the statistical challenge of exploring a parameter space with a dimension of several thousand, as well as the computational demand of factorizing a large matrix at each MCMC step, the estimation of the signal covariance parameters between genes was not performed jointly. Instead, each pair of genes was fitted separately, with a single-channel Gaussian Process being first used to estimate the signal variance and bandwidth parameters for each gene and this estimate being used as a prior for the (pairwise) joint inference. This procedure effectively breaks down a Gaussian Process inference of any size into several smaller inference problems requiring factorization of a matrix of size 2*N*, with a total number of parameters of the order of *N*, which are computationally much more manageable and can be run in parallel. Because the covariance parameters depend only on the relationship between two variables (here, genes), separate estimation does not affect inference of the parameters; in fact, it removes the constraint of positive-definiteness on the matrix of covariances of all genes (which instead applies to the matrix of two genes only, see S1 Appendix.

Eight parallel chains were run for each estimation with 40 thousand samples each; half were excluded as warm-up and 1 out of every 40 was kept for further calculations. Convergence was assessed using the $\hat{R}$ metric and observing the number of effective samples (ESS) [75]. The annotated model implemented in the Stan probabilistic language is made available at https://github.com/caesoma/Multiple-shifts-in-gene-network-interactions-shape-phenotypes-of-Drosophila-melanogaster. Because inference was done separately for each selection scheme, differences between them were assessed by comparing the posterior distribution of the parameters of interest.

## Confirmatory experiments

We tested *Minos* insertions putatively disrupting five genes from the significant Gaussian Process correlation for their effect on sleep, along with their two background controls. We assayed phenotypes using the same procedure outlined above. Twenty-four flies per sex per line were assayed, and the experiment was replicated twice. Sleep was analyzed using the following ANOVA model:

$$Y = \mu + Genotype + Sex + Rep + Genotype \times Sex + Genotype \times Rep$$
$$+ Sex \times Rep + Genotype \times Sex \times Rep + \varepsilon$$

where *Sex* and *Rep* are as previously defined above and *Genotype* refers to the *Minos* insertion

line or control. We conducted RNA-Seq on the *Minos* insertion lines and their controls. We collected 10 flies per sex/line at the conclusion of sleep monitoring for RNA extraction. RNA was extracted as detailed above, with the following exception: ERCC spike-ins (ThermoFisher Scientific, Waltham, MA) were added to the RNA after the extraction procedure. A total of 28 samples (7 lines × 2 sexes × 2 replicates) were collected and processed. We then sequenced these samples and processed them as detailed above. Note that we discarded one sample due to failed quality control during library preparation. We compared normalized gene expression in the mutants to their respective controls using a Kruskal-Wallis non-parametric test. Expression ratios were computed between knock down genotypes and their wild-type controls ($w^{1118}$ or $y^1 w^{67c23}$) for individual genes predicted by the Gaussian Processes to be significantly correlated with the knocked down genes in the relevant sex-selection scheme combination (S1 Table), henceforth candidate genes. Expression ratios were also computed for sets of 1000 genes chosen at random, each set matching the genetic backgrounds (knockdown and controls) as well as the sex (henceforth random sets); a distribution of expression ratios was generated for each random set.

## Results

### Phenotypic response to artificial selection

The selection procedure for night sleep was very effective. Long-sleeper and short-sleeper populations had significant differences in night sleep across all generations ($P_{Sel}$ = 0.0003; S2 Table); in fact, night sleep was different for the two selection schemes for each generation considered separately except for generations 0 and 1 (S3 Table). Both males and females responded equally to the selection procedure. Fig 1A shows the phenotypic response to 13 generations of selection for night sleep. At generation 13, the long-sleeper populations averaged 642.2 ± 3.83 and 667.8 ± 2.97 minutes of night sleep for Replicate 1 and Replicate 2, respectively. The short-sleeper populations averaged 104.3 ± 6.71 and 156.2 ± 8.76 minutes of night sleep for Replicate 1 and Replicate 2, respectively. The average difference between the long- and short-sleeper lines was 537.9 minutes for Replicate 1, and 511.6 minutes for Replicate 2. In contrast, the two control populations did not have differences in their night sleep after 13 generations of random mating ($P_{Gen}$ = 0.7083; S4 Table). In the initial generation, night sleep was 519.6 ± 10.57 minutes in the Replicate 1 control and 567.9 ± 7.63 minutes in the Replicate 2 control. At generation 13, night sleep was 563.4 ± 7.62 and 542.3 ± 7.91 in Replicates 1 and 2, respectively, a difference of only 43.8 and 25.6 minutes. These negligible changes in night sleep in the control population suggest that little inbreeding depression occurred over the course of the experiment [59]. Selection was asymmetric, with a greater phenotypic response in the direction of reduced night sleep. Note also that night sleep is bounded from 0 to 720 minutes, and the initial generation had 515.39 minutes of night sleep on average across all populations, a fairly long night sleep phenotype. This high initial sleep may explain why the response to selection for short night sleep was more effective. Night sleep is sexually dimorphic [11, 83, 84]; yet both males and females responded to the selection protocol equally ($P_{Sel \times Sex}$ = 0.9492; S2 Table). Thus, we constructed a set of selection populations with nearly 9 hours difference in night sleep.

In an artificial selection experiment, some amount of inbreeding will necessarily take place. Only a subset of the animals are selected each generation as parents; thus phenotypic variance is expected to decrease as selection proceeds [59]. However, this is not the case for all artificial selection experiments [59]. We calculated the coefficient of environmental variation ($CV_E$) [58] and evaluated its trajectory across time in order to determine whether the populations were becoming more or less variable over time. As Fig 1B shows, night sleep $CV_E$ increased

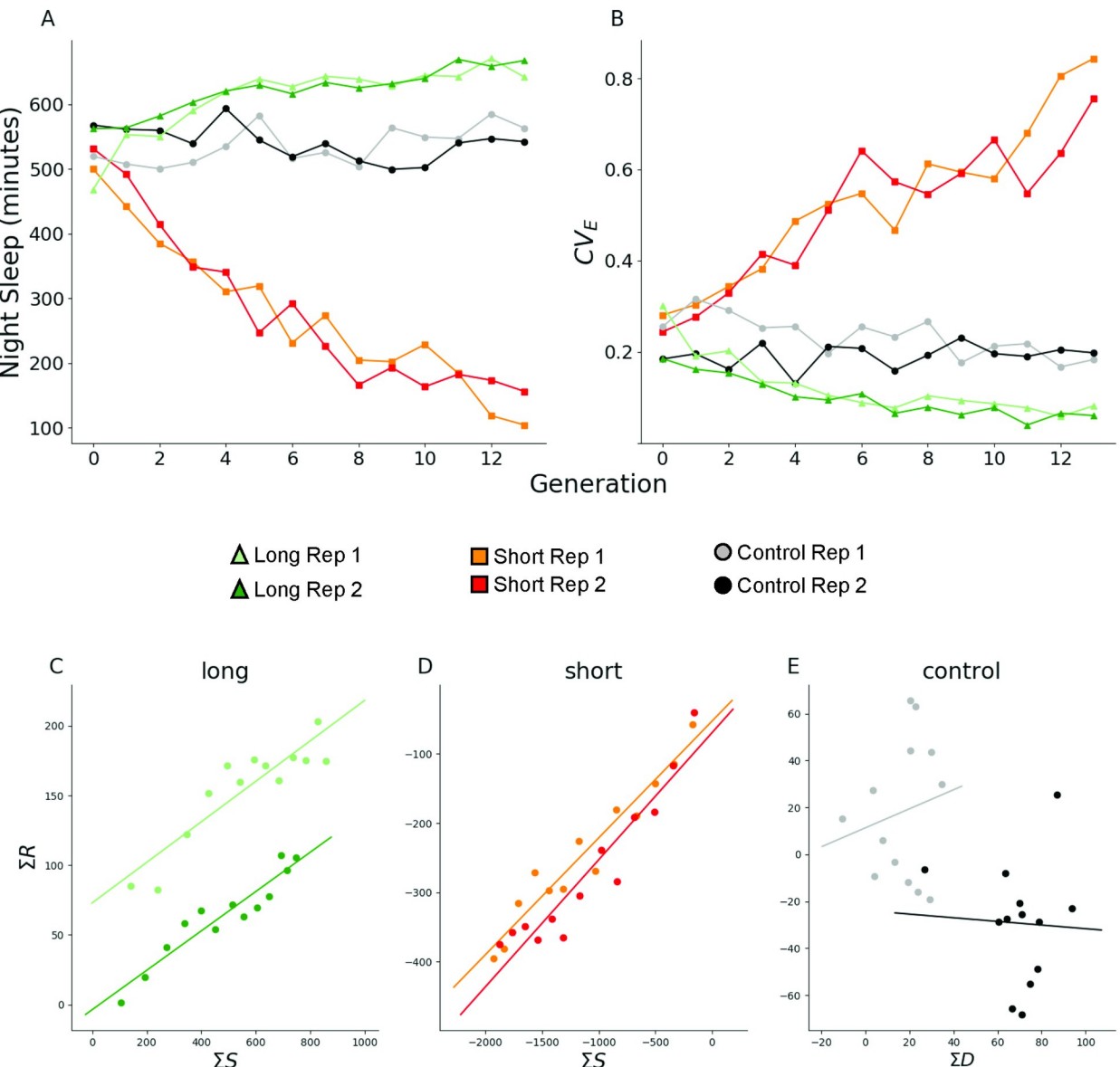

**Fig 1. Response to artificial selection for night sleep.** (A) Mean and (B) coefficient of environmental variation of night sleep. Plot and regression lines of cumulated selection differential (ΣS) against cumulated selection response (ΣR) for (C) long- and (D) short-sleeping populations, and against cumulated differential ΣD for (E) controls. Light green, Replicate 1 long-sleeper population; Dark green, Replicate 2 long-sleeper population; Orange, Replicate 1 short-sleeper population; Red, Replicate 2 short-sleeper population; Gray, Replicate 1 control population; Black, Replicate 2 control population.

over time in the short sleepers, and decreased over time in the long sleepers ($P < 0.0001$; S5 Table). The increase in $CV_E$ in short sleepers was largely due to a decrease in the population mean as the standard deviation also decreased over time, indicating that the phenotypic variance decreased (S2 Fig). Likewise, the standard deviation decreased in the long sleepers over time, even as the mean night sleep increased, indicating decreased variability in these populations as well. These changes in $CV_E$ mimic previous observations in populations artificially selected for sleep [10]. Regressions of the cumulated response on the cumulated selection differential were used to estimate heritability ($h^2$). Long-sleeper population $h^2$ (±*SE* of the

coefficient of regression) were estimated as 0.145 ± 0.021 and 0.141 ± 0.014 (all *P* < 0.0001) for Replicates 1 and 2, respectively (Fig 1C); short-sleeper population $h^2$ were 0.169 ± 0.013 and 0.183 ± 0.019 (all *P* < 0.0001) for Replicates 1 and 2 (Fig 1D). In contrast, estimated regression coefficients for the control population were non-significant and with high standard errors associated to the regression estimates: 0.405 ± 0.695 (*P* = 0.57) and −0.078 ± 0.487 (*P* = 0.88) for Replicates 1 and 2, respectively (Fig 1E).

## Correlated response of other sleep traits to selection for night sleep

Traits that are genetically correlated with night sleep might also respond to selection for long or short night sleep [59]. Indeed, some sleep and activity traits have been previously shown to be phenotypically and genetically correlated [11, 83, 84]. We examined the other sleep and activity traits for evidence of a correlated response to selection. Night and day average bout length (*P* = 0.0008 and *P* = 0.0391, respectively) and sleep latency (*P* = 0.0023) exhibited a correlated response to selection for night sleep across generations 0–13, while night and day bout number, day sleep, and waking activity did not (S2 Fig; S2 Table). In the case of day average bout length, the correlated response was sex-specific to males (*P* = 0.0140) (S2 Table). Significant correlated responses for night and day average bout length and sleep latency did not occur in all generations (S3 Table). Night average bout length responded to selection for night sleep in most generations, while day average bout length responded in only four of the last six generations. Sleep latency responded to selection after the second generation. In addition, we observed significant differences between the long-sleeping and short-sleeping populations for the $CV_E$ of all sleep traits except waking activity $CV_E$ (S2 Fig; S5 Table). However, the pattern of the $CV_E$ for each trait appeared to be more random across time. These correlated responses concur with previous observations we made in selected populations originating from the same outbred population for night sleep and night average bout length, and night sleep and sleep latency [10]. However, unlike the previous study, we did not see a correlated response between night sleep and day sleep, and night sleep and day bout number [10]. The lack of correlated response reflects the relatively low genetic correlation these two traits have with night sleep [11, 84].

## Phenotypes in flies used for RNA-Seq

Every generation, we harvested RNA from flies chosen at random from the 200 measured for sleep in each selection population, with the exception of the flies chosen as parents for the next generation. We extracted RNA from two replicates of 10 flies each per sex and selection population. Since these flies amount to only 20% of the flies measured for sleep each generation, their sleep may or may not be representative of the group as a whole. We therefore correlated the mean night sleep for each generation in the flies harvested for RNA with the mean night sleep of all flies measured to determine how similar night sleep was to the total in the group (S3 Fig). The correlations were very high for the selected populations: long-sleeper flies harvested for RNA were very well correlated with the total measured in each population [$r^2$ = 0.99 and 0.96 (all *P* < 0.0001) for Replicate 1 and 2 respectively], as were short-sleepers [$r^2$ = 0.99 for Replicate 1 and 0.97 for Replicate 2 (all *P* < 0.0001)]. The control populations, which did not undergo selection, were somewhat less well correlated. Replicate 1 of the control population had an $r^2$ of 0.75 (*P* = 0.0001) and Replicate 2 had an $r^2$ of 0.85 (*P* < 0.0001). The lower correlations observed in control flies indicate that they were less inbred than the selected populations. Thus, the flies harvested for RNA are very good representatives of each population as a whole.

## Hierarchical Generalized Linear Model analysis reveals that selection for night sleep impacts gene expression

For each gene, the linear model analysis produced posterior distributions for the parameters as well as log-likelihood values for the full and reduced models. Point estimates (MAP) are shown in S6 Table for females and S7 Table for males. For the male flies 11,778 genes passed the filtering for low expression, of which 405 were found to have a significant selection scheme effect over the generations of artificial selection (i.e., significant likelihood ratio test for the *sel × gen* term). Thus, the expression level shift given by the slope of the generalized linear model is different from controls and attributable to selection for long and/or short sleep. For the females 820 genes out of 9,370 with detectable expression were found to be significant. Genes with opposite trends in the short and long selection schemes were compared using the group-level parameter $\mu_{short \times gen}$ and $\mu_{long \times gen}$ (i.e. the effect that best explains both replicates): 384 genes in females (S8 Table) and 204 genes in the males (S9 Table) showed opposite trends by that criterion. Between males and females, 85 genes were common to both sexes. Known functions of these 85 genes from the DAVID gene ontology database are presented in S10 Table. We used these 85 genes in subsequent analyses; see below. Fig 2 shows the fit for one gene.

## Pairwise Spearman correlation is non-specific and significant for a large fraction of genes

We computed Spearman correlations for all pairwise combinations of the 85 genes common between sexes (S11 Table). Correlations computed using the Spearman method were found to be significant at 95% confidence for 2,999 of the 3,570 possible pairs. The confidence intervals for the correlation coefficients showed no overlap with controls for either short sleepers, long sleepers, or both populations in 1,348 of 3,570 pairs. Thus, a simple correlational analysis identifies a minimum of 38% of the possible interactions among genes as relevant.

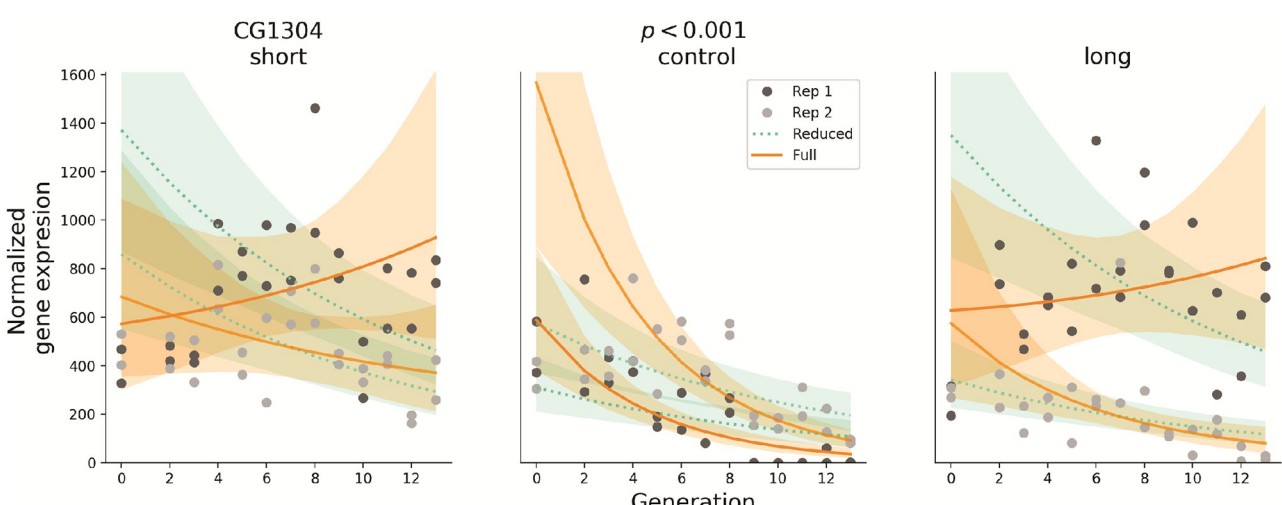

**Fig 2. Fit of Hierarchical Generalized Linear Model to gene *CG1304* for flies selected for short sleep, unselected controls, and selected for long sleep.** The solid lines show the expected value of full model, dashed lines for reduced model, and shaded regions show the 95% credibility interval. Replicate 1 data points are shown in dark gray, Replicate 2 in light gray.

## Gaussian Process model analysis uncovers nonlinear trends and specifically identifies covariance in expression between genes

As noted above, a simple correlational analysis suggested that large numbers of genes are potentially interacting to alter sleep. Because direct computation of linear model-based correlations cannot account for non-linear effects or spurious confounding trends we fit Gaussian Process models that can account for temporal variation in multiple genes even in the absence of actual interactions between them. The 85 significant genes overlapping between males and females potentially have 3,570 pairwise interactions. To that end, the parameter of interest in the Gaussian Process model is the signal covariance between each pair of genes. This covariance is a measure of the degree of their interaction. We applied the Gaussian Process model for each of the 3,570 pairs for each selection scheme (long, short, and control). As an example, the model fit for one pair of genes from the female gene expression data is shown in Fig 3.

Convergence for all three runs was on the order of $|\hat{R} - 1| \approx 10^{-4}$, and close to the 4,000 samples expected for each run; therefore, the wide confidence intervals are likely a product of the large dispersion in the data itself. Correlation between gene expression patterns of the two genes is computed by dividing the signal covariance by the square root of the signal variance of each gene—e.g. $\rho_l = \sigma^2_{l(ij)}/\sigma_{l(i)}\sigma_{l(j)} = \sigma^2_{long(LysC,CG1304)}/\sigma_{long(LysC)}\sigma_{long(CG1304)}$—that is, similar to

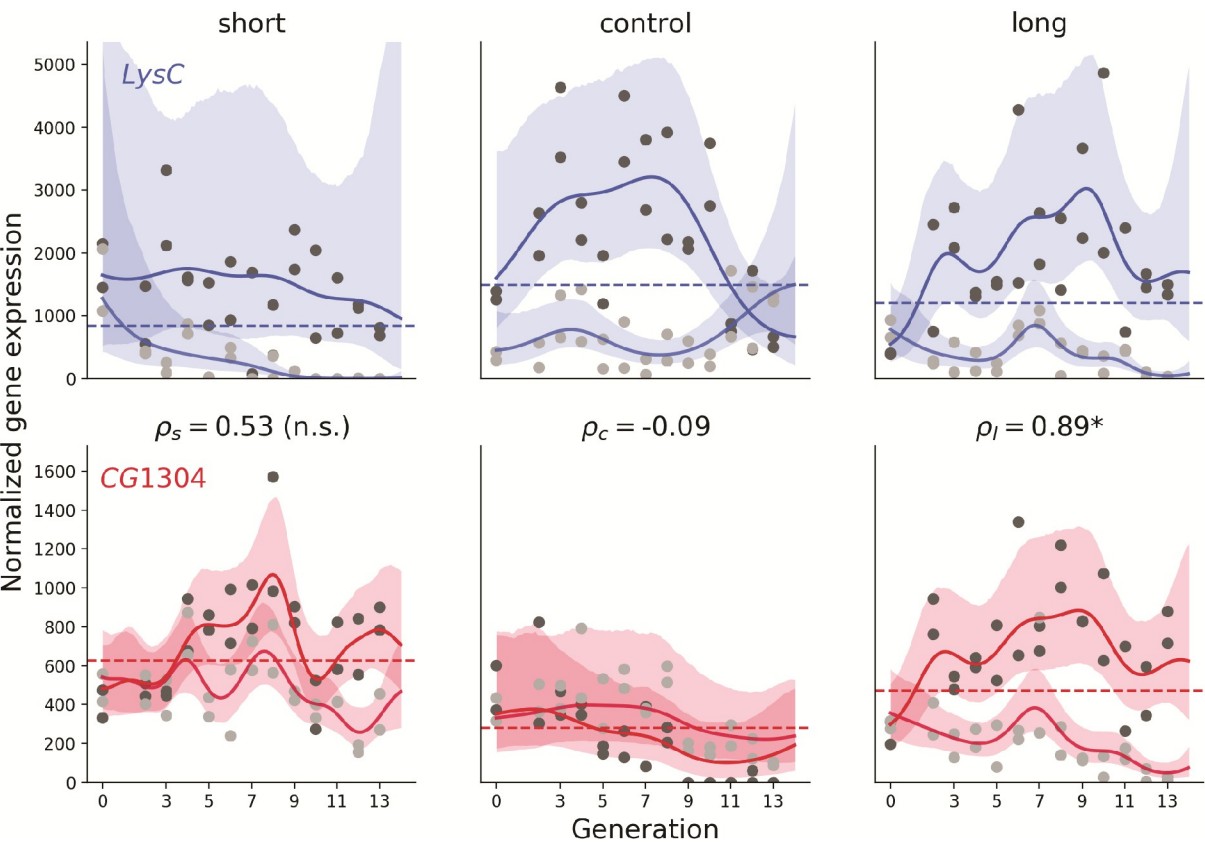

**Fig 3. Fit of Gaussian Process model to pair of genes *LysC* and *CG1304*, for female flies selected for short sleep, unselected controls, and selected for long sleep.** The solid lines show the expected value, while the shaded regions show the 95% credibility interval. Replicate 1 data points are shown in dark gray, Replicate 2 in light gray). The expectation for correlations ($\rho_{sel}$) is shown for each selection scheme. An asterisk indicates significant difference from controls in selection scheme, as opposed to non-significance (n.s.).

computing a correlation coefficient from variances and covariances, but taken as the expectation over the posterior distribution obtained from MCMC.

Fig 3 illustrates the nonlinear trajectories of gene expression that cannot be detected by the GLM model. The two trajectories exhibited high signal covariance between the expression of the two genes in the long sleepers ($\rho_l = 0.89$) that was significantly different from controls; however, intermediate covariance in the short sleepers ($\rho_s = 0.53$) did overlap with that of controls, and therefore was not significantly different.

S4 Fig part A shows a pair where interactions in both short and long selection schemes are different from controls, S4 Fig part B shows another pair of genes where neither scheme is different from controls. This illustrates a range of possibilities, including a case where Spearman correlations are significant but GP correlations are not (the opposite also occurs). Parts C and D of S4 Fig fit each gene individually, and the fit does not change substantially between single to multiple channel models.

The 85 single-channel fits were good despite varying levels of dispersion and occasional outliers, indicating no issues with the Gaussian Processes' ability to fit the temporal patterns of any one gene. For the two-channel inference, upwards of 90% of the chains initially converged under the criterion that $0.95 < \hat{R} < 1.05$; because the inference method is stochastic it is expected that by chance some chains may not converge and/or mix well with their replicates. Chains that initially failed were rerun up to two times. After three runs over 99% of the chains converged; the reasons for lack of convergence of the remaining were not investigated further. Fig 4 shows six heat maps (one for each sex and selection scheme combination) with the

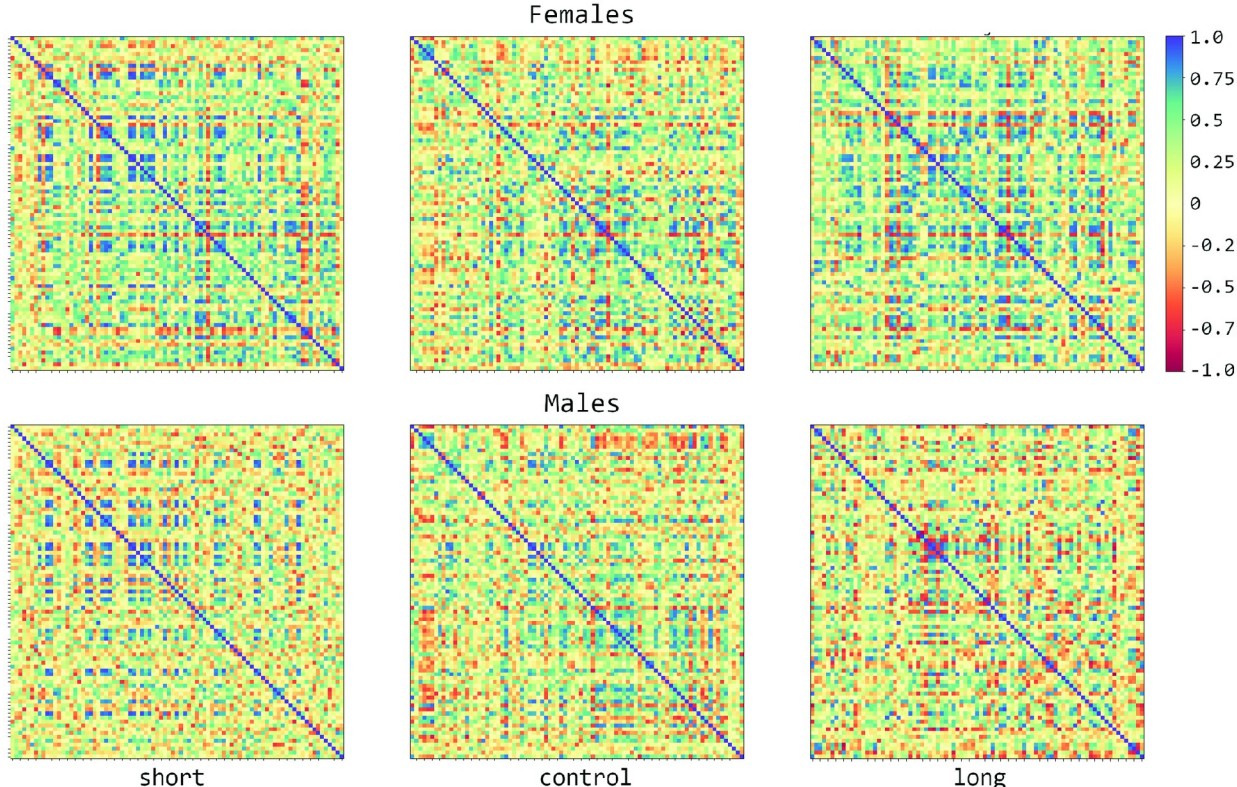

**Fig 4. Signal variances and covariances normalized to range [-1,1] for females and males in each of the selection schemes: Short, control, and long.** Each off-diagonal square is the expected value of the interaction between two of 85 genes, for a total of 3,570 pairs.

correlations for all pairs of genes calculated as described in the previous figure, summarizing the inferred interactions. Of the 3,570 correlations, 1,612 were greater than 0.5 and 98 greater than 0.9.

In addition to computing expected values, the posterior distributions were used to compare the signal covariances between selection schemes and set a cutoff. Distributions of the parameter for each sex-selection scheme were assembled from the parallel MCMC runs; 145 gene pairs in the selected populations are found to be different from controls (i.e. do not overlap with them at 95% credibility for either short, long or both populations). Out of the 145, twelve gene pairs were common between males and females selected for long night sleep and one pair to males and females selected for short sleep; one gene pair was common to females in both selection schemes, and three pairs were common to males. S11 Table shows the expected values of signal covariances normalized by the variances for all two-way interactions side by side with the Spearman correlations. S12 Table shows the subset of significant Gaussian Processes correlations.

We constructed a network for each sex/selection scheme combination based on the magnitude of the correlation between genes. The network for males selected for long sleep having significant gene interactions is shown in Fig 5 (S5 Fig shows the networks for the remaining three sex-selection scheme combinations). S13 Table lists the number of connections (degrees) that each gene has with the others in the network. The average number of connections for long-sleeper males was 2.6; the other three networks had average degrees of 2.0 or less (2.0 for long-sleeper females and short-sleeper males; 1.75 for short-sleeper females).

For comparison, looking at significant ($\rho_{sel} \neq 0$) Spearman correlations keeps almost three thousand interactions (i.e. excludes just a bit more than a tenth of the genes), and comparing the distributions $\rho_{sel}$ versus $\rho_{control}$—similar to how the Gaussian Processes are compared—still has over thirteen hundred. Therefore, computing correlations between genes using covariance estimates from the Gaussian Processes appears to increase specificity over direct correlations. Furthermore, the Gaussian Processes appear to be more sensitive in finding 68 gene pairs that are not found to be significant by the first Spearman approach and 18 not found by the second.

Finally, we examined known interactions between the 85 genes and any other genes using the *Drosophila* Interaction Database, DroID [85]. We found 2,830 interactions; 8 of these were one of the 3,570 between the 85 genes, but none of them overlapped with the 145 gene pairs found to be different from controls. The gene interactions we observed may therefore be unique to sleep.

## Mutational analyses confirms role of candidate genes and interacting gene expression networks in sleep

We tested five genes for differences in sleep as compared to their isogenic control: *CG12560*, *CG13793*, *Cytochrome P450 6a16* (*Cyp6a16*), *highwire* (*hiw*), and *Jonah 65Aii* (*Jon65Aii*) (S1 Table). All of the *Minos* insertions altered night sleep (Fig 6). Night sleep increased from 50—115 minutes beyond the $w^{1118}$ control line (all *P*-values < 0.0125, the Bonferroni-corrected *P*-value; S1 Table). Flies having a *Minos* insertion in *Jon65Aii* slept 66 minutes less than their corresponding $y^1 w^{67c23}$ control (*P* < 0.0001). All *Minos* insertions had the same directional effect on night sleep for both males and females, but only the *CG12560* and *Jon65Aii* insertions had statistically significant effects on night sleep on each sex separately (S1 Table). Thus, all genes affected night sleep duration, but *CG12560* and *Jon65Aii* had the greatest effect on both sexes.

Gene expression decreased significantly in the *CG12560* and *Jon65Aii Minos* insertions relative to their controls (S6 Fig). The remaining *Minos* insertion lines had some changes in gene

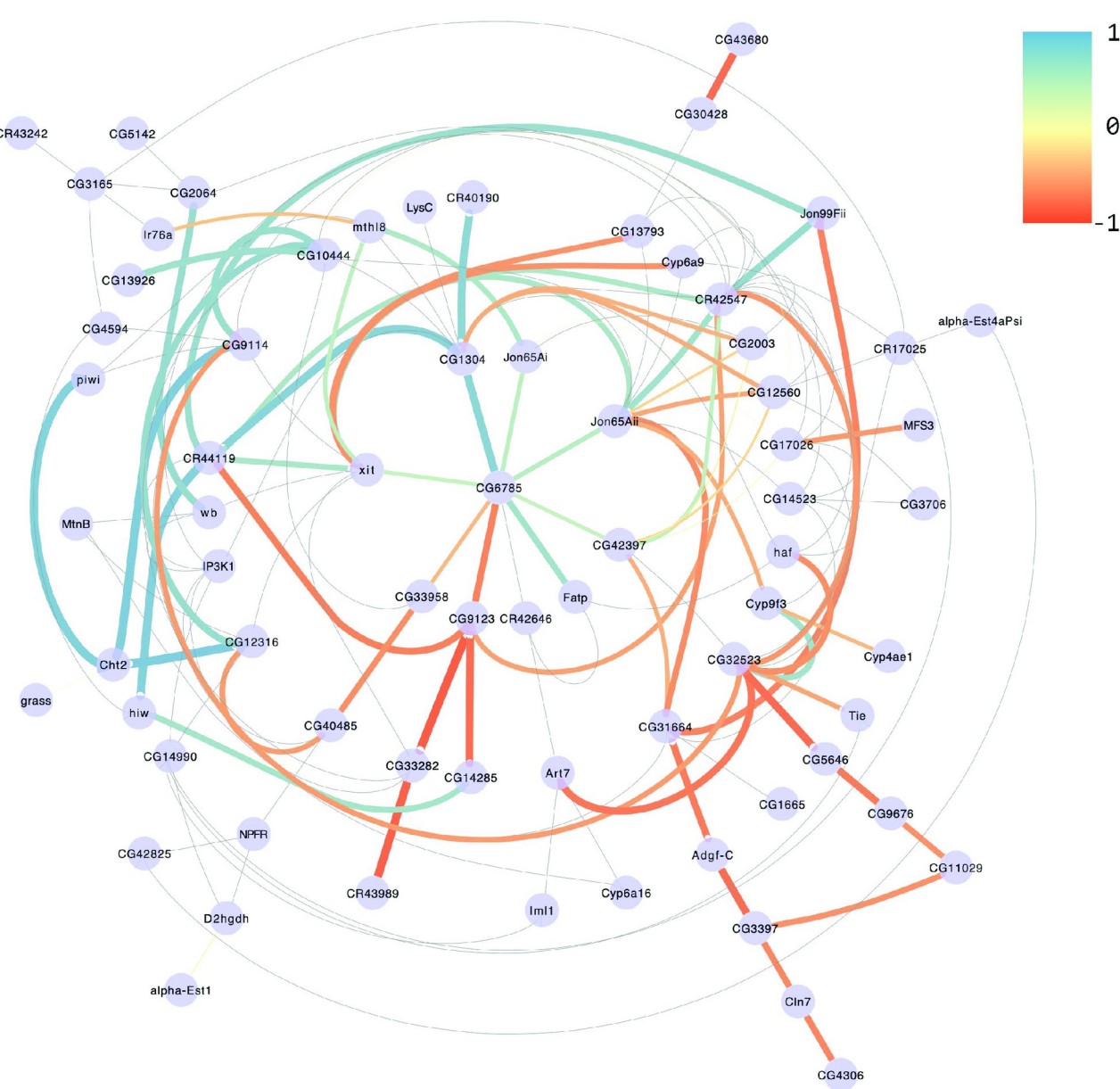

**Fig 5. Gene interaction network in males selected for long sleep.** Edges represent signal covariances whose posterior distributions do not overlap with that of controls at 95% credibility. Colors and line thickness indicate indicate the strength and the direction of the correlation. Thin gray lines show all 145 interactions significant for at least one of the four sex-selection scheme combinations.

expression relative to the control, however, the changes were not formally significant. Potential reasons for the lack of a significant change in gene expression in the remaining lines include: the position of the insertion within the targeted gene, which has variable effects on its expression; the relatively low statistical power of the experiment; confining our observation to a single timepoint during the day; or pooling whole flies, which might obscure gene expression changes occurring at a single-tissue level.

Our baseline expectation was that expression levels between knockdown and control lines should not be affected for most genes. For candidate genes, we hypothesized that the ratio of

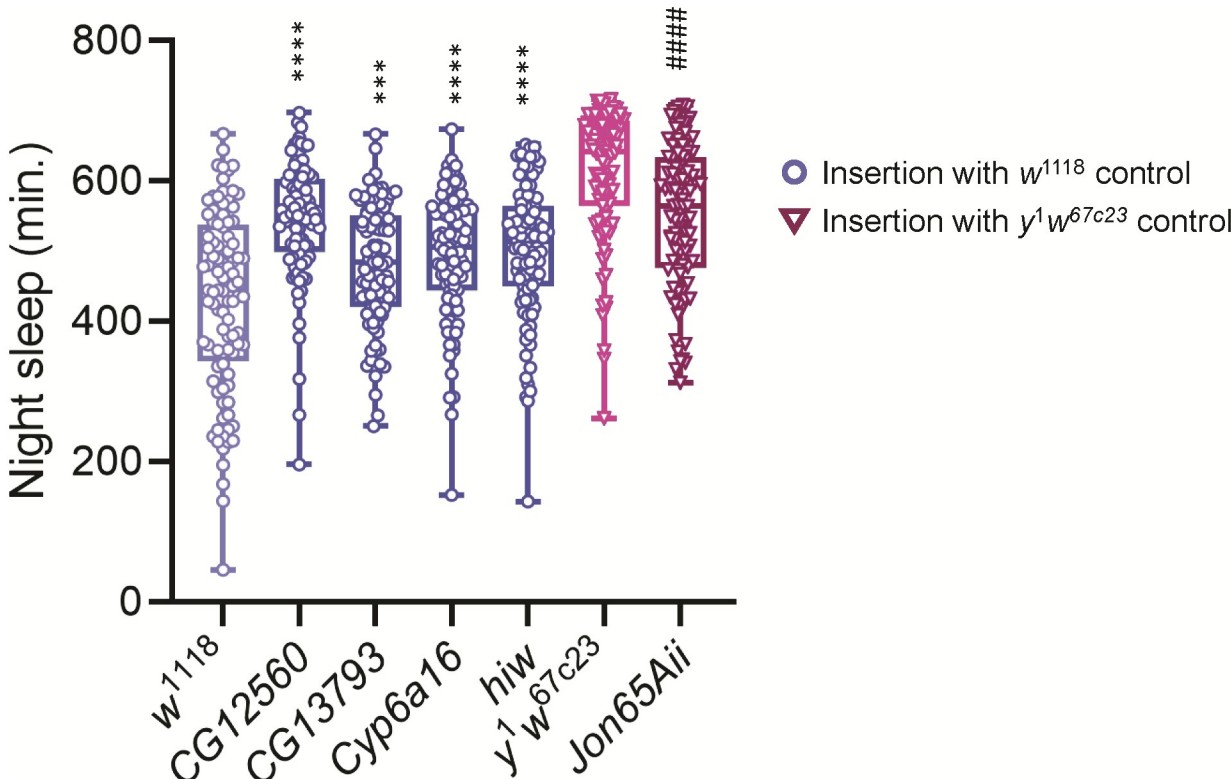

**Fig 6. Night sleep in *Minos* mutations.** The figure compares night sleep duration in each mutant with that of its respective control. Blue circles indicate *Minos* insertions with $w^{1118}$ control strain; red triangles indicate *Minos* insertions with $y^1w^{67c23}$ control strain. **** or ####, *P* < 0.0001; ***, *P* < 0.001, three-way ANOVA. All *P*-values are less than the Bonferroni-corrected *P*-value of 0.0125.

gene expression between the *Minos* insertion line and its respective control would differ considerably from 1.0; conversely, the ratio of gene expression between *Minos* insertion line and control would be approximately 1.0 for unrelated genes—that expectation is confirmed by the distributions consistently centered around unity (median = 0.995). We plotted the ratios of candidate genes against the distributions of the matching random sets (Fig 7). Our supposition was largely realized for *CG12560* and *Jon65Aii*, the two genes having significant knockdown in gene expression (S7 Fig and S14 Table).

## Discussion

We have shown that robust, reproducible phenotypic changes in *Drosophila melanogaster* sleep are associated with hundreds (405 in males, 820 in females) of individual shifts in gene expression—and as a consequence hundreds of thousands of potential combinations $[\binom{405}{2} > 8 \cdot 10^4$ and $\binom{820}{2} > 3 \cdot 10^5]$. Nevertheless, unique interactions important to the phenotypes are a comparatively small number (145 out of $\binom{85}{2} = 3570$ possible combinations of the 85 genes common to males and females). We have also shown that these interactions cannot be found with linear model analyses or conventional correlation calculations only, but are specifically identified using a combination of an informative experimental design with densely-sampled time points to generate a large scale data set, and a nonparametric, nonlinear model-based approach that explicitly accounts for covariance in gene expression.

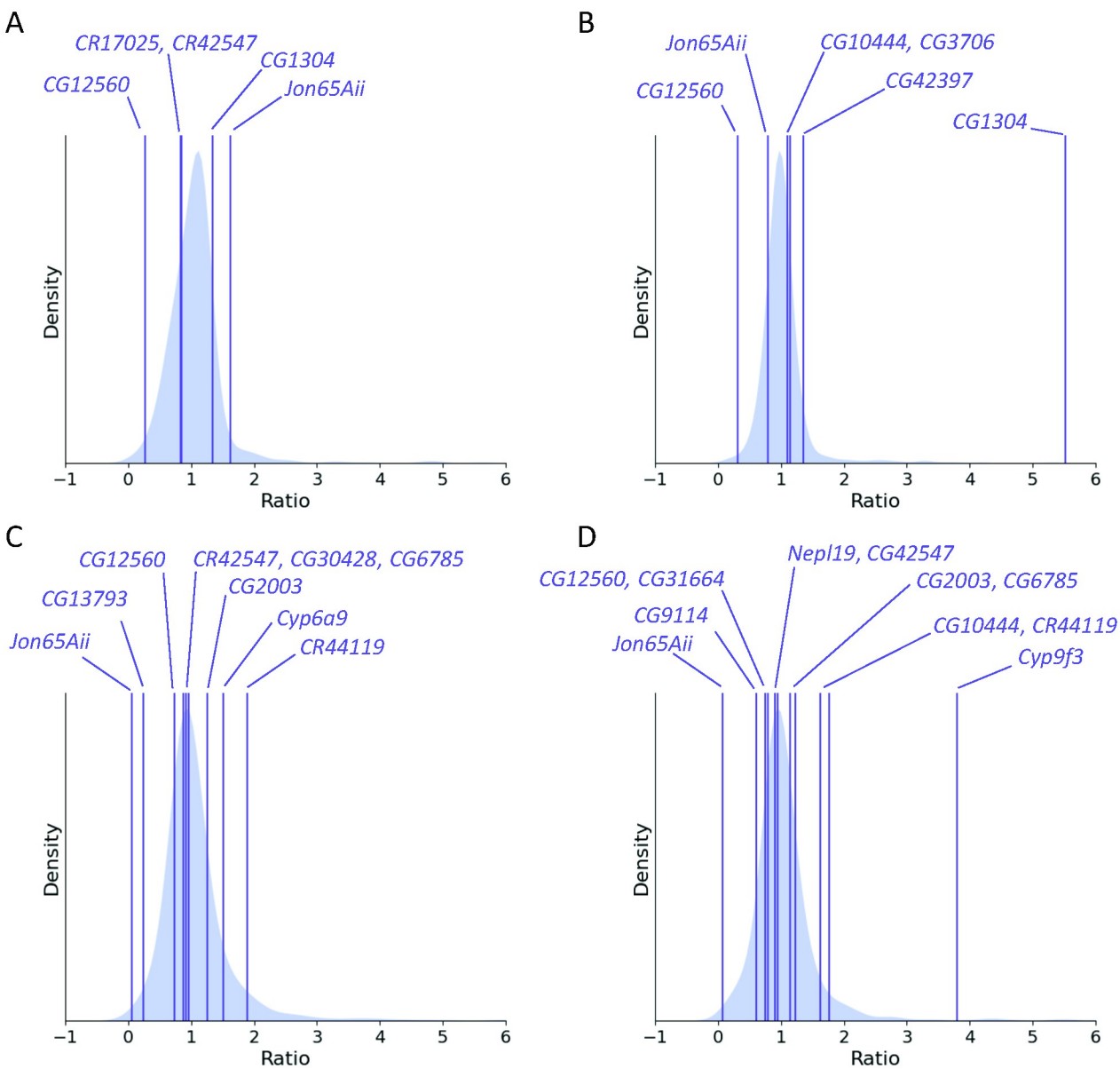

**Fig 7. Comparison of ratios of gene expression between genes with significant Gaussian Process correlations and unrelated genes for *CG12560* and *Jon65Aii* mutants.** Purple lines show the ratio of mutant gene expression to control for genes with significant Gaussian Process correlations. The distribution of gene ratios for 1,000 unrelated genes is plotted in the background. Genes having the most extreme ratios are indicated; see S14 Table for the calculations. (A) *CG12560* females; (B) *CG12560* males; (C) *Jon65Aii* females; (D) *Jon65Aii* males.

The genes we identify herein overlap and extend previous work. Of the 1,140 genes implicated in the generalized linear model, 151 (13.2%) overlapped with previous candidate gene, random mutagenesis, gene expression, and genome-wide association studies of sleep and circadian behavior in flies [10, 11, 84, 86–99]. Notably, previous studies identified the genes *CG17574*, *cry*, *dro*, *mip120*, *Mtk*, *NPFR1*, *pdgy*, *PGRP-LC*, *Shal*, and *vari* as affecting sleep duration [84, 88–91, 93, 97, 99]. Two genes, *ringer* and *mip120*, overlapped with our previous study of DNA sequence variation in flies selected for long and short sleep [10]. In that study we identified a polymorphism in an intron of *ringer* that changed in allele frequency with selection,

with increases in the population frequency of the 'G' allele with increasing sleep, while the frequency of the 'A' allele increased with decreasing sleep. When the selective breeding procedure was relaxed, the frequency of the 'G' allele increased in short-sleeping populations, paralleling an increase in sleep [100]. One possibility is that this polymorphism contributes to the changes in gene expression in *ringer* that we observed in the present study. Of the 85 genes common to both sexes that we used in the gene interaction networks, 11 (13%) appear in other studies of sleep: *CG10444*, *CG2003*, *CG5142*, *CG6785*, *CG9114*, *CG9676*, *CR42646*, *hiw*, *NPFR1*, *Tie*, and *wb* [11, 89, 92, 95]. Thus, our study corroborates genes known to affect sleep, and identifies new candidate genes for sleep as well.

Interestingly, our Gene Ontology analysis identified nine genes from the 85-gene network with predicted Serine endopeptidase/peptidase/hydrolase activity: *CG1304*, *CG10472*, *CG14990*, *CG32523*, *CG9676*, *grass*, *Jon65Ai*, *Jon65Aii*, and *Jon99Fii*. All of these genes are expressed in neurons and epithelial cells, and all genes are expressed at the adult stage [101]. Serine proteases are a large group of proteins (257 in Drosophila) that perform a variety of functions [102]. Their predicted enzymatic activity suggests a putative role in proteolysis. This is an intriguing observation given pioneering work in mammals which suggested a role for sleep in exchanging interstitial fluid and metabolites between the brain and cerebral spinal fluid [8]. Recent work demonstrated that a similar function is conserved in flies via vesicular trafficking through the fly blood-brain barrier [103]. It would be interesting to determine whether these genes function in this process.

We observed changes in night sleep duration for all *Minos* insertions tested, and the effects were more prominent in females. Sex-specific effects of mutations on sleep are common in flies [84, 104–108], and sex-biased effects are often noted in females [109–113]. Remarkably, we noted gene expression relationships among genes with predicted significant Gaussian Process correlations in the *Minos* insertion lines despite the fact that the sleep was neither extremely long or short in mutants or controls, and the genetic background of these lines ($w^{1118}$ or $y^1 w^{67c23}$) is completely different from the outbred Sleep Advanced Intercross Population that we used for artificial selection.

Here we extracted RNA from flies at a single circadian timepoint, ZT6. However, gene expression is known to cycle in fly heads and bodies, begging the question of whether the genes we identified at a single timepoint are subject to cycling over the 24-hour day. We therefore compared our list of genes that were significantly associated with selection scheme over generation (405 genes for males; 820 genes for females; 85 genes overlapping both sexes) with genes known to have cycling expression [114–120]. We found that 47 of the 405 genes identified for males cycle (11.6%), 170 of the 820 genes identified for females cycle (20.7%), and 13 of the 85 genes overlapping between males and females cycle (15.3%). Thus, most of the genes we identified are not known to cycle over the 24-hour day.

That complex traits can be mostly explained by additive effects of individual genes (and their expression) is a common and sometimes useful assumption. While it underpins preliminary analyses that allow whole-transcriptome data to be understood, it eliminates the ability to infer interactions between them from the data and stops short from identifying relevant processes. Complex traits involve multiple genes, and the actual interactions giving rise to phenotypes are likely to be highly nonlinear [121]. These nonlinearities are not a mathematical construct, but a biological reality arising from chemical kinetics. Favoring approaches that account for these features will not only increase statistical power, but understanding of actual biological mechanisms beyond simple network representations of gene expression [122].

In most correlation and information-theory based methods the dimension (e.g. time or space) across which samples covary is only implicit [36]; the only possible conclusion from a significant correlation between two sets of observations is that one may have an effect on the

other—i.e. the data alone does not allow the distinction between actual interactions and spurious correlation. Bioinformatic pipelines that have correlation as their starting point—in addition to carrying over its limitations—are not straightforwardly comparable to our approach (see S1 Appendix). In the context of Gaussian Processes, correlation between all pairs of data points—including within the same time series, i.e. autocorrelation—is explicit in time (or other dimension), so similar trends do not necessarily imply covariance between the sets of observations. Therefore, on the one hand GPs are a nonparametric method that requires no more biological knowledge than that for computing a linear correlation; on the other hand, while not an explicit description of dynamic biological processes, it is also a model-based approach that can be used within more mechanistic formalisms like differential equations [41, 43], or potentially be used to formulate specific hypotheses and build mechanistic models.

Although somewhat self-evident, it is important to highlight the fact that to describe correlations along time, multiple time points are needed—put another way, the use of a nonlinear model requires enough resolution in the data that the trajectory can be identified. To that end, a single high-resolution, large data set with a specific design, like the one generated in this work, will be more useful than several small data sets, for instance with only initial and final time points and allowing only two-sample linear comparison. Gene expression measured at the terminal generation of selection and compared among selected and control groups does identify candidate genes [24, 25, 28, 29, 31–33], but the relationship between pairs of genes is lost. Some studies evaluated gene expression during the last 2–3 generations of selection [27, 30]; however, the additional sampling was used to confirm consistency rather than change across time. Our approach of sampling over time enabled us to derive interactions between genes and demonstrated that unique gene expression network profiles develop in long sleepers as compared to short sleepers.

When employing methods of increasing complexity or sophistication there is always the question of how relevant the inference is or, in other words, how "real" are the parameters or processes in the model. This pursuit of simplicity may favor the use of methods based on linear models as more palpable approaches and less prone to arbitrary assumptions about how the parameters are put together; however, it is important to realize that linear coefficients are no more real than those of any other model. On the contrary, biological processes are not restricted by our ability to comprehend them. Therefore, what may seem as an Occam's Razor-like simplicity will probably hinder accurate description of nature. Systems-level understanding of complex biology requires not only more and more detailed data, but better descriptions of the processes and methodology that captures higher-order phenomena. Equivalently, experimental validation of these phenomena will be more technically challenging to accomplish. Despite the additional difficulties, it must be recognized that methods that cannot possibly match the complexity of nature are doomed to scratch all over the surface without realizing a deeper understanding. The Gaussian Processes we apply herein have broad applications to other experimental designs, such as gene expression measured at varying time intervals over the circadian day, or time-based sampling of gene expression responses to drug administration.

## Supporting information

**S1 Fig. Principal Component Analysis (PCA).** PCA on matrix of normalized expression data shows complete separation of sexes along the first component, which explains 65% of the variance in the data.
(PDF)

**S2 Fig. Correlated response to selection for long/short night sleep and associated coefficient of environmental variation.** A, day bout number; B, day bout number coefficient of

environmental variation ($CV_E$); C, day sleep; D, day sleep $CV_E$; E, night bout number; F, night bout number $CV_E$; G, night sleep; H, night sleep $CV_E$; I, waking activity; J, waking activity $CV_E$; K, sleep latency; L, sleep latency $CV_E$; M, day average bout length; N, day average bout length $CV_E$; O, night average bout length; P, night average bout length $CV_E$; Q, night sleep standard deviation. Light green, Replicate 1 long-sleeper population; Dark green, Replicate 2 long-sleeper population; Orange, Replicate 1 short-sleeper population; Red, Replicate 2 short-sleeper population; Gray, Replicate 1 control population; Black, Replicate 2 control population. $CV_E$, coefficient of environmental variation.
(PDF)

**S3 Fig. Correlation of night sleep between flies harvested for RNA and all flies in the population.** A, long-sleeping Replicate 1; B, long-sleeping Replicate 2; C, short-sleeping Replicate 1; D, short-sleeping Replicate 2; E, control Replicate 1; F, control Replicate 2.
(PDF)

**S4 Fig. Gaussian Process model fits to selected genes.** A, fit of Gaussian Process model to pair of genes *haf* and *CG1304*; B, fit of Gaussian Process model to pair of genes *CR43242* and *CG1304*; C, fit of single-channel Gaussian Process model to *CG1304* gene; D, fit of single-channel Gaussian Process model to *LysC* gene.
(PDF)

**S5 Fig. Gene interaction networks.** A, Males selected for short sleep; B, Females selected for long sleep; C, Females selected for short sleep.
(PDF)

**S6 Fig. Gene expression in *Minos* mutants.** For each candidate gene, the gene expression in the *Minos* mutant and corresponding control are plotted. $^*$ or $^{\#}P < 0.05$ by Kruskal-Wallis test. A, *CG12560*; B, *Jon65Aii*; C, *CG13793*; D, *Cyp6a16*; E, *hiw*.
(PDF)

**S7 Fig. Comparison of ratios of gene expression between genes with significant Gaussian Process correlations and unrelated genes for *CG13793*, *Cyp6a16*, and *hiw Minos* mutants.** A, *CG13793* females; B, *CG13793* males; C, *Cyp6a16* males; D, *hiw* females; E, *hiw* males.
(PDF)

**S1 Appendix. Notes on multichannel Gaussian Processes.**
(PDF)

**S1 Table. Effects of *Minos* insertions on sleep.** For each gene the table lists the Flybase ID, Bloomington *Drosophila* Stock Center (BDSC) number, *Minos* genotype, and isogenic control line. For each sleep trait, the number of flies tested and mean sleep phenotype is given for sexes combined and females and males separately. *P*-values are listed for each term in the ANOVA model for sexes combined and for males and females separately. Significance is indicated by bold *P*-values.
(XLSX)

**S2 Table. Quantitative genetics of the response to selection for long or short night sleep and related sleep parameters.** For each trait, the ANOVA analysis results are presented. Source indicates each factor in the model. *gen*, generation; *rep*, replicate; *sel*, selection scheme; *d.f.*, degrees of freedom; M.S., Type III mean squares; F, F ratio statistic; P, P–value.
(XLSX)

**S3 Table. Quantitative genetics of the response to selection for long or short night sleep per generation.** For each sleep trait, the ANOVA analysis results are presented for each generation. Source indicates each factor in the model. *rep*, replicate; *sel*, selection scheme; *d.f.*, degrees of freedom; M.S., Type III mean squares; *F*, *F* ratio statistic; *P*, *P*-value.
(XLSX)

**S4 Table. Quantitative genetics of control populations.** For each sleep trait, the ANOVA analysis results are presented. Source indicates each factor in the model. *gen*, generation; *rep*, replicate; *d.f.*, degrees of freedom; MS, Type III mean squares; *F*, *F* ratio statistic; *P*, *P*-value.
(XLSX)

**S5 Table. Correlated response of sleep trait coefficient of environmental variance ($CV_E$) to selection for long or short night sleep duration.** For each sleep trait listed, the ANOVA results are presented. Source indicates each factor in the model. *gen*, generation; *sel*, selection scheme; *d.f.*, degrees of freedom; M.S., Type III mean squares; *F*, *F* ratio statistic; *P*, *P*-value.
(XLSX)

**S6 Table. GLM analysis results for females.** GLM analysis results for each gene in females are shown as a row; the Maximum a Posteriori (MAP) parameter estimates and log-likelihoods are shown as well as *p*-values computed from the likelihood ratio test. Significance statistics corrected for multiple testing are also included, as well as the normalized counts for all samples.
(XLSX)

**S7 Table. GLM analysis results for males.** GLM analysis results for each gene in males are shown as a row; the Maximum a Posteriori (MAP) parameter estimates and log-likelihoods are shown as well as *p*-values computed from the likelihood ratio test. Significance statistics corrected for multiple testing are also included, as well as the normalized counts for all samples.
(XLSX)

**S8 Table. Genes with opposite slopes for the short and long interaction terms of generation in females.** Columns have the same meaning as those in S6 Table.
(XLSX)

**S9 Table. Genes with opposite slopes for the short and long interaction terms of generation in males.** Columns have the same meaning as those in S7 Table.
(XLSX)

**S10 Table. Gene Ontology (GO) analysis results for 85 significant genes common to males and females.** The table lists GO classification (Biological Process (BP), Molecular Function (MF), or Cellular Component (CC)); the GO term description; the number of genes associated with each GO term and their percentage relative to the total number of genes with that GO term in *D. melanogaster*; the enrichment *P* value, and the Benjamini-adjusted *P* value.
(XLSX)

**S11 Table. Correlations obtained from normalizing Gaussian Process signal covariances (GP correlation) and from Spearman Correlation for each of the six sex and selection scheme combinations.**
(XLSX)

**S12 Table. Expected values for the correlations obtained from normalizing Gaussian Process signal covariances (GP correlation) that do not overlap with controls for each of the**

**six sex and selection scheme combinations.** The value is missing if there is an overlap with controls in that condition.
(XLSX)

**S13 Table. Degree for each gene in the GP network.** For each sex and selection scheme, the table lists the number of genes connected to the gene in the network. NA, not applicable.
(XLSX)

**S14 Table. Gene expression ratios calculated from *Minos* insertion line and control RNA-Seq data.** For each target gene the table lists the corresponding control line, the Flybase ID and gene symbol of the gene predicted to interact with the target gene, the normalized expression of the interacting gene for the *Minos* target gene line and isogenic control line, and the ratio of normalized expression (*Minos*/control).
(XLSX)

**S15 Table. Night sleep phenotypes.** For each selection scheme, sex, generation, and population replicate, the number of flies, mean night sleep, and standard deviation (SD) of night sleep are listed.
(XLSX)

## Acknowledgments

We thank the members of the NISC Consortium for sequence data and helpful discussions. This work used the computational resources of the National Institutes of Health High-Performance Computing Biowulf cluster (http://hpc.nih.gov). We thank N. Redekar and N. Gulzar from the NIAID Collaborative Bioinformatics Resource (NCBR) for assistance with bioinformatic processing.

## Author Contributions

**Conceptualization:** Caetano Souto-Maior, Susan T. Harbison.

**Data curation:** Caetano Souto-Maior.

**Formal analysis:** Caetano Souto-Maior, Susan T. Harbison.

**Funding acquisition:** Susan T. Harbison.

**Investigation:** Caetano Souto-Maior, Yazmin L. Serrano Negron.

**Methodology:** Caetano Souto-Maior.

**Project administration:** Susan T. Harbison.

**Visualization:** Caetano Souto-Maior, Yazmin L. Serrano Negron, Susan T. Harbison.

**Writing – original draft:** Caetano Souto-Maior, Yazmin L. Serrano Negron, Susan T. Harbison.

**Writing – review & editing:** Caetano Souto-Maior, Susan T. Harbison.

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
