## [Decision Letter · Decision Letter 0]

13 Apr 2023

Dear Dr. Harbison,

Thank you very much for submitting your manuscript "Nonlinear expression patterns and multiple shifts in gene network interactions underlie robust phenotypic change in Drosophila melanogaster selected for night sleep duration" for consideration at PLOS Computational Biology.

As with all papers reviewed by the journal, your manuscript was reviewed by members of the editorial board and by several independent reviewers. In light of the reviews (below this email), we would like to invite the resubmission of a significantly-revised version that takes into account the reviewers' comments.

We cannot make any decision about publication until we have seen the revised manuscript and your response to the reviewers' comments. Your revised manuscript is also likely to be sent to reviewers for further evaluation.

Sincerely,

Attila Csikász-Nagy

Academic Editor

PLOS Computational Biology

James O'Dwyer

Section Editor

PLOS Computational Biology

Reviewer's Responses to Questions

**Comments to the Authors:**

Reviewer #1: • What are the main claims of the paper and how significant are they for the discipline?

In this manuscript the authors developed a Gaussian Process model to address the limitation of the current methods in capturing non-linear covariance between genes over time. Given how the assumption of linearity is expected to be often violated in complex biological systems, this is a significant methodological advance into exploring and describing complex interactions bio-chemical interactions, for example, and as illustrated, to study associations of transcript-phenotype during artificial selection.

• Are these claims novel? If not, which published articles weaken the claims of originality of this one?

Thought the statistical and mathematical principles of the methodology developed have been studied and applied in other fields, to the best of my knowledge, it is the first time such methodologies are used to explore the selection of gene networks during selective breeding for extreme sleep phenotype.

• Are the claims properly placed in the context of the previous literature? Have the authors treated the literature fairly?

This was one of the points raised by the previous reviewers and the authors have addressed it by providing a more complete discussion of the previous biological knowledge.

• Do the data and analyses fully support the claims? If not, what other evidence is required?

The data supports the authors’ claims. In addition to the computational aspect of the work, and despite it being impossible to confirm all results from high through-put computational modeling, the authors present relevant biological evidence in support of their findings.

• Would additional work improve the paper? How much better would the paper be if this work were performed and how difficult would it be to do this work?

These issue has been raised by the previous reviewers and addressed properly in my opinion.

• PLOS Computational Biology encourages authors to publish detailed protocols and algorithms as supporting information online. Do any particular methods used in the manuscript warrant such treatment? If a protocol is already provided, for example for a randomized controlled trial, are there any important deviations from it? If so, have the authors explained adequately why the deviations occurred?

The underlying mathematical principles of this work have been described, though not in sleep research literature. As such a detailed description of the algorithms behind it is not necessary.

• Are original data deposited in appropriate repositories and accession/version numbers provided for genes, proteins, mutants, diseases, etc.?

The sequencing data has been made available on the National Center for Biotechnology Information (NCBI) Gene Expression Omnibus (GEO) under the accession number GSE202600.

• Has the author-generated code that underpins the findings been made publicly available?

The code used has been made available on GitHub.

• Does the study conform to any relevant guidelines such as CONSORT, MIAME, QUORUM, STROBE, and the Fort Lauderdale agreement?

As far as I can tell the study conforms with all standards.

• Are details of the methodology sufficient to allow the experiments to be reproduced?

The information provided in the methods and online should be enough to reproduced the manuscript, though it may be difficult for non-experts to use it without a walk through or tutorial guide.

• Is any software created by the authors freely available?

Yes, all code has been made available on GitHub

• Is the manuscript well organized and written clearly enough to be accessible to non-specialists?

The manuscript is well organized and easy t follow, though non-specialists lacking the mathematical background required may not find it accessible.

• Does the paper use standardized scientific nomenclature and abbreviations? If not, are these explained at the first usage?

All nomenclature is standard or properly explained.

• Additional points.

For the figures, it could be helpful to add the color legend to the side, so one doesn’t need to read the text every time to find out which phenotype is which color.

Since it was discussed with the previous reviewers, it could be worth mentioned a little on ZT6 (‘middle of the day’) fits in the fly sleep-wake cycle, since their behavior differs from other animal models.

For the section ‘Phenotypes in flies used for RNA-seq’ , I would intuitively expect that the unselected control flies would be more correlated with the population they were drawn from than the selected groups (short- and long-sleepers). This is mostly due to the fact that the selected populations are expected to be the phenotypical outliers, so the furthest away from the mean behavior. Could the authors comment or clarify these observations?

Reviewer #2: Summary

In the manuscript entitled “Nonlinear expression patterns and multiple shifts in gene network interactions underlie robust phenotypic change in Drosophila melanogaster selected for night sleep duration”, the authors used a Gaussian Process (GP) model to systematically identify changes in interactions between genes over the generations of artificial selection. Specifically, they conducted an artificial selection experiment with Drosophila melanogaster, and, every generation, they harvested RNA from files. Then, they established a multi-channel GP model for estimation of the nonlinear expression patterns of genes related to sleep phenotypic change, which is impossible with a linear model analysis as mentioned in the manuscript. Using the GP model, they identified some genes whose expression level significantly changes in the artificial selection. Lastly, to validate part of the findings obtained using the GP model, mutational analyses were performed by a Minos insertion. These results clearly demonstrate the usefulness of the GP-based approach in investigating the complex molecular interactions leading to diverse phenotypes. I believe that the approach is interesting and correctly applied. Considering this, the current manuscript well fits the scope of the PLOS Computational Biology, and thus I recommend the publication. However, I provide major and minor comments to be solved before the publication in below.

Major comments

1. To understand this work, readers must have both experimental and mathematical/statistical backgrounds, as Reviewer 1 mentioned. However, several potential readers of this work, such as evolutionary biologists and clinician in sleep research field might not have enough mathematical/statistical backgrounds. I believe that this problem frequently occurs in interdisciplinary research. I felt that, to circumvent this problem, the authors described their methodology in detail in the Methods and Materials, which is good. However, even with the detailed description, the manuscript can lose potential readers due to the inherent difficulties of the methodology, making readers a little bit stressful. To completely avoid this circumstance, I kindly recommend that the authors provide a user-friendly computer package to implement their GP algorithm. Here, user-friendly means that users do not need to understand all the code, and, by just changing the input (e.g., RNA-seq data), they can run the code and analyze their own data. After this, it would be great if the authors clearly mention that they provide an easy-to-use and user-friendly computer package for implementation of the algorithm, for example, in the introduction section.

2. Approaches based on GP models have been recently used to infer dynamics of complex biological systems. Specifically, Yang et al. (Yang et al. PNAS 2021 PMID: 33837150) proposed a method called MAGI that use a Gaussian process model over time series data. It would be great if the authors can clearly describe advantages (or novelty) of the method proposed in the manuscript compared to the MAGI in the manuscript.

3. Because the authors said “we develop a flexible Gaussian Process model …” in the abstract, and the manuscript focuses on the application of the method to experimental data, I believe that the authors need to describe previous approaches of network analysis potentially related to the proposed GP method (e.g., Langfelder & Horvath BMC Bioinformatics 2008 PMID: 19114008; Yang et al. PNAS 2021 PMID: 33837150; Horvath & Dong PLOS Comp, Biol. 2008 PMID: 18704157) in the introduction section. Moreover, it would be great if the authors can more clearly describe the novelty of this work. Specifically, if there is a methodological advance in the proposed method compared to the previous ones, please clearly describe it. If the main contribution of the manuscript is a specific application of the previously developed GP method and identification of previously unknown biological knowledge, please more focus on describing the biological findings instead of describing the adopted methodology.

Minor comments

1. As the author described in the Methods and Materials, 10 files were chosen from 75 files and frozen for RNA extraction at ZT6. Among the extracted RNAs, levels of some RNA might oscillate over time, as the Review 2 mentioned. Thus, the conclusion of the manuscript may change depending on the collection time of RNA. It would be great if the authors discuss this point in the discussion section in addition to mentioning the collection time in the Methods and Materials. At least, it might be needed to mention that the future work that checks whether the findings are preserved even with the samples collected at different times in the manuscript.

2. I could not access the Git Hub repository via the link provided in the Data Availability section. It would be great if the authors can check whether the link works.

3. I believe that this technique can be applied to other biological systems, such as gene regulatory networks associated with Fatty acid Composition (Basnet et al. Plant Physiol. 2016 PMID: 26518343) and cell signaling networks associated with antibiotic stress (Kim et al. Science Advances 2022 PMID: 35302852). It would be great if the authors mention this point in the discussion section.

**Have the authors made all data and (if applicable) computational code underlying the findings in their manuscript fully available?**

Reviewer #1: Yes

Reviewer #2: **No: **I could not access the computer codes via the Git Hub repository link provided in the Data Availability section. This might be due to the problems in my side. I am not sure...

PLOS authors have the option to publish the peer review history of their article (what does this mean?). If published, this will include your full peer review and any attached files.

Reviewer #1: No

Reviewer #2: No
---

## [Decision Letter · Decision Letter 1]

27 Jun 2023

Dear Dr. Harbison,

Thank you very much for submitting your manuscript "Nonlinear expression patterns and multiple shifts in gene network interactions underlie robust phenotypic change in Drosophila melanogaster selected for night sleep duration" for consideration at PLOS Computational Biology. As with all papers reviewed by the journal, your manuscript was reviewed by members of the editorial board and by several independent reviewers. The reviewers appreciated the attention to an important topic. Based on the reviews, we are likely to accept this manuscript for publication, providing that you modify the manuscript according to the review recommendations.

Address the remaining concern of the referee by expanding on related methods and positioning your method better among them.

Sincerely,

Attila Csikász-Nagy

Academic Editor

PLOS Computational Biology

James O'Dwyer

Section Editor

PLOS Computational Biology

Reviewer's Responses to Questions

**Comments to the Authors:**

Reviewer #2: I appreciate the thorough response of the authors to my comments, which effectively addressed several of my concerns. Nevertheless, there are still one remaining concern that should be addressed before the manuscript can be published in PLoS Computational Biology. See below for details.

Concern 1 (regarding the third major comment):

The authors developed a novel GP model for analyzing nonlinear expression patterns in gene network interactions and applied it to biological data. I acknowledge their methodological advancement in this regard. However, it is important to ensure that the authors have clearly described their approach in relation to existing literature and have treated the literature fairly, as emphasized by previous reviewers.

Typically, when introducing a new methodological advancement in a manuscript, it is necessary to carefully discuss and contextualize previous methods in the introduction and discussion sections. The authors have made improvements to the discussion section based on my comments about MAGI, which is commendable. However, they have not adequately introduced any previous methods in the introduction section. It appears that the authors believe that mentioning the previous literature about GP models in the supplementary information (i.e., appendix) is sufficient. I am not sure whether this approach is enough to treat the previous literature fairly, especially considering that one of the main contributions of the manuscript is the development of a new GP model.

To the best of my knowledge, when the authors introduced their methodology in the introduction section, they merely stated, "The RNA sequence data, which consisted of expression levels as a function of time (measured in generations), was analyzed using a Multi-Channel Gaussian Process [37, 38]." I am unsure if this single sentence adequately conveys to readers that the authors have developed a new method distinct from previous approaches in the manuscript.

In conclusion, I believe that the findings presented by the authors meet the high-quality standards required for publication in PLoS Computational Biology. However, significant improvements are needed in the writing to address the aforementioned concerns.

**Have the authors made all data and (if applicable) computational code underlying the findings in their manuscript fully available?**

Reviewer #2: Yes

PLOS authors have the option to publish the peer review history of their article (what does this mean?). If published, this will include your full peer review and any attached files.

Reviewer #2: No

Figure Files:

Data Requirements:

Reproducibility:

References:

---

## [Editor Report · Decision Letter 2]

25 Jul 2023

Dear Dr. Harbison,

We are pleased to inform you that your manuscript 'Nonlinear expression patterns and multiple shifts in gene network interactions underlie robust phenotypic change in Drosophila melanogaster selected for night sleep duration' has been provisionally accepted for publication in PLOS Computational Biology.

Best regards,

Attila Csikász-Nagy

Academic Editor

PLOS Computational Biology

James O'Dwyer

Section Editor

PLOS Computational Biology

---

## [Editor Report · Acceptance letter]

4 Aug 2023

PCOMPBIOL-D-23-00247R2 

Nonlinear expression patterns and multiple shifts in gene network interactions underlie robust phenotypic change in *Drosophila melanogaster* selected for night sleep duration

Dear Dr Harbison,

I am pleased to inform you that your manuscript has been formally accepted for publication in PLOS Computational Biology. Your manuscript is now with our production department and you will be notified of the publication date in due course.

With kind regards,

Zsofia Freund
